# Socio-Economic Benefits of *Colophospermum mopane* in a Changing Climate in Northern Namibia

**Andreas Nikodemus *** , **Azadeh Abdollahnejad** , **Alpo Kapuka** , **Dimitrios Panagiotidis and Miroslav Hájek**

Faculty of Forestry and Wood Sciences, Czech University of Life Sciences Prague, Kamýcká 129, Praha 6-Suchdol, 16500 Prague, Czech Republic
* Correspondence: nikodemus@fld.czu.cz

**Abstract:** Millions of local communities in southern Africa depend on forest ecosystems and the goods and services they provide for their livelihoods. This paper aims to assess the socio-economic benefits of forest goods and services in a changing climate by focusing on the forest products of *Colophospermum mopane* (*C. mopane*) in the Kunene and Omusati regions in northern Namibia. We used *C. mopane* product data from 2011 to 2021. Our analyses showed that local communities harvested five main products from *C. mopane*, namely firewood, poles, droppers, rafters, and roots. Firewood and poles were the primary *C. mopane* products harvested by local communities, mainly for subsistence use. Our results suggest that *C. mopane* potentially continues to the provision of goods and services for the livelihood of local communities, despite the changing climate in northern Namibia. We propose future studies in predictive analysis focus on extreme weather events, such as forest fires, droughts, floods, and other climate-related hazards that affect goods and services provided by forest ecosystems in the northern regions and the entire country.

**Keywords:** climate change; *Colophospermum mopane*; commercial use; forest products; Kunene region; Omusati region; rural communities; subsistence use; northern Namibia

## 1. Introduction

Millions of local communities in southern Africa depend on forest ecosystems and the goods and services they provide for their livelihoods [1,2], including ecosystem services provided by *C. mopane* (Kirk ex Benth) species. In addition to services such as carbon sequestration and the protection of watersheds and biodiversity [3], forest ecosystems provide goods and services to local communities in various forms in the region, ranging from timber and non-timber forest products (NTFPs) [4]. Local communities utilize forest products, such as fuel wood, construction materials, medicine, and food, for marketing and household consumption [5]. However, various previous studies have acknowledged that forest ecosystems in southern Africa are highly vulnerable to climate change [1,2,6]. Some of the identified climate risks to forest ecosystems in the region include, for example, altering the growth rates of woodland flora and impacting species composition and productivity [3,7].

The southern part of Africa is one of the most vulnerable regions to climate change [8]. This situation is due to the region's high exposure to climate change, poor socio-economic conditions, increased reliance on natural resources, and inability to implement adaptive measures effectively [8–10]. For example, in the Sub-Saharan African region, unpredictable rainfall and recurrent droughts significantly impact agricultural production [11], aggravating consequences, including high risks to food security [12].

Climate change projections for southern Africa further predict increased droughts, the frequency, and intensity of wildfires, land degradation, low agricultural and vegetation productivity, extreme temperatures, and increased food insecurity [13]. Furthermore, climate change is also associated with increasing desertification [14]. Additionally, according

to [15], extreme temperatures, high evapotranspiration, and high human activity intensity are some of the main drivers of desertification in some parts of Africa, particularly southern Africa.

Namibia is considered the driest country in southern Africa [16–20]. As a result, Namibia is a southern African country that is significantly impacted by climate change due to its arid conditions [21–23]. Climate change has been a significant challenge, threatening progress toward the country's national and millennium development goals [24]. The recent prolonged drought experienced in the country has severely impacted the functions of both human communities and their livelihoods, as well as ecological ecosystems, particularly in providing essential ecosystem services [21,24].

Namibia's temperature trends have risen by 0.58 °C to 1.2 °C on average over the past 50 years, with more significant increases in the country's northern parts [25]. Rainfall trends are less evident than temperature trends over the past 50 years, and there are substantial variations in the direction and magnitude of the changes observed across the region [25,26]. Precipitation trends are challenging to discern given the country's typically erratic rainfall; extreme rainfall contributes a significant proportion of the annual rainfall in some regions [25,27]. However, there is no adequate research concerning long-term climate variability across the northern Namibia regions to date [15].

There is substantial literature about the impacts of climate change on various forest and woodland species in southern Africa [28,29], including studies addressing multiple aspects of *C. mopane* [30–34]. However, studies focusing on the socio-economic benefits of mopane woodlands in a changing climate in southern Africa, particularly in northern Namibia, are still limited [35–38]. Therefore, this paper aims to assess the socio-economic benefits of *C. mopane* in a changing climate by focusing on the products harvested for use at the local community level.

## 2. Materials and Methods

### 2.1. Study Area

The geographical location of Namibia is in the south-west of Africa. South Africa borders it to the east and south-east, Zambia to the north-east, Angola to the north, and Botswana to the east [22]. It is a semi-dry country with deficient rainfall compared to other countries in the southern part of Africa [21]. Due to its unique climate conditions, Namibia is considered one of the most vulnerable countries to climate change impacts [13]. Namibia has a population of 2.5 million and covers a total surface area of 824,292 km$^2$ [24].

We focused on two of the fourteen political regions of Namibia (the Omusati and Kunene regions) [39] (Figure 1). The two target regions fall in the northern geographical zones of Namibia [40]. Together, they cover 17% of the total area and constitute 13% of the population of Namibia. The target regions also form part of the *Baikiaea-mopane* woodlands of southern Africa [41] and represent areas with the highest distribution of *C. mopane* species in Namibia, with the Omusati region having the highest distribution of the species between the two regions [42].

The northern part of Namibia is predominantly rural and is one of the hotspots for climate-change-related impacts [24,43]. Rural communities in the target regions heavily rely on subsistence agricultural practices and other forest-related ecosystem services, particularly from the *Baikiaea-mopane* woodlands, for their livelihoods [44,45]. Agricultural practices alone contribute between 22% and 32% of the local livelihoods of the Omusati and Kunene regions, respectively [46–48]. Between 2001 and 2018, the two regions experienced a combined tree cover loss of 302 hectares [49].

The area is semi-arid and characterized by high temperatures, ranging from 5 °C to 37 °C, whereas the annual average rainfall is about 350-500 mm between November and April [50]. The yearly rainfall of the Omusati region ranges from 400 mm to 500 mm per annum [48]. Average daily temperatures vary from 6 °C to 35 °C depending on the season. For example, summers are sweltering, with a maximum temperature of between 30 °C and 35 °C during the hottest months [46]. The Kunene region occupies the north-west corner of

Namibia and shares borders with the Omusati region in the eastern part [51]. Therefore, the two regions have similar climatic conditions. However, slight differences in weather and climatic conditions are noticeable. Irregular annual rainfall increases from the west to the east of the region from less than 50 mm to 415 mm [51,52]. Average daily temperatures range from 5 °C to 35 °C depending on the season [47]. Summer day temperatures are often sweltering, reaching up to 35 °C with minimum temperatures of 14 °C on average. During the winter, temperatures can range from 5 °C to 26 °C [47].

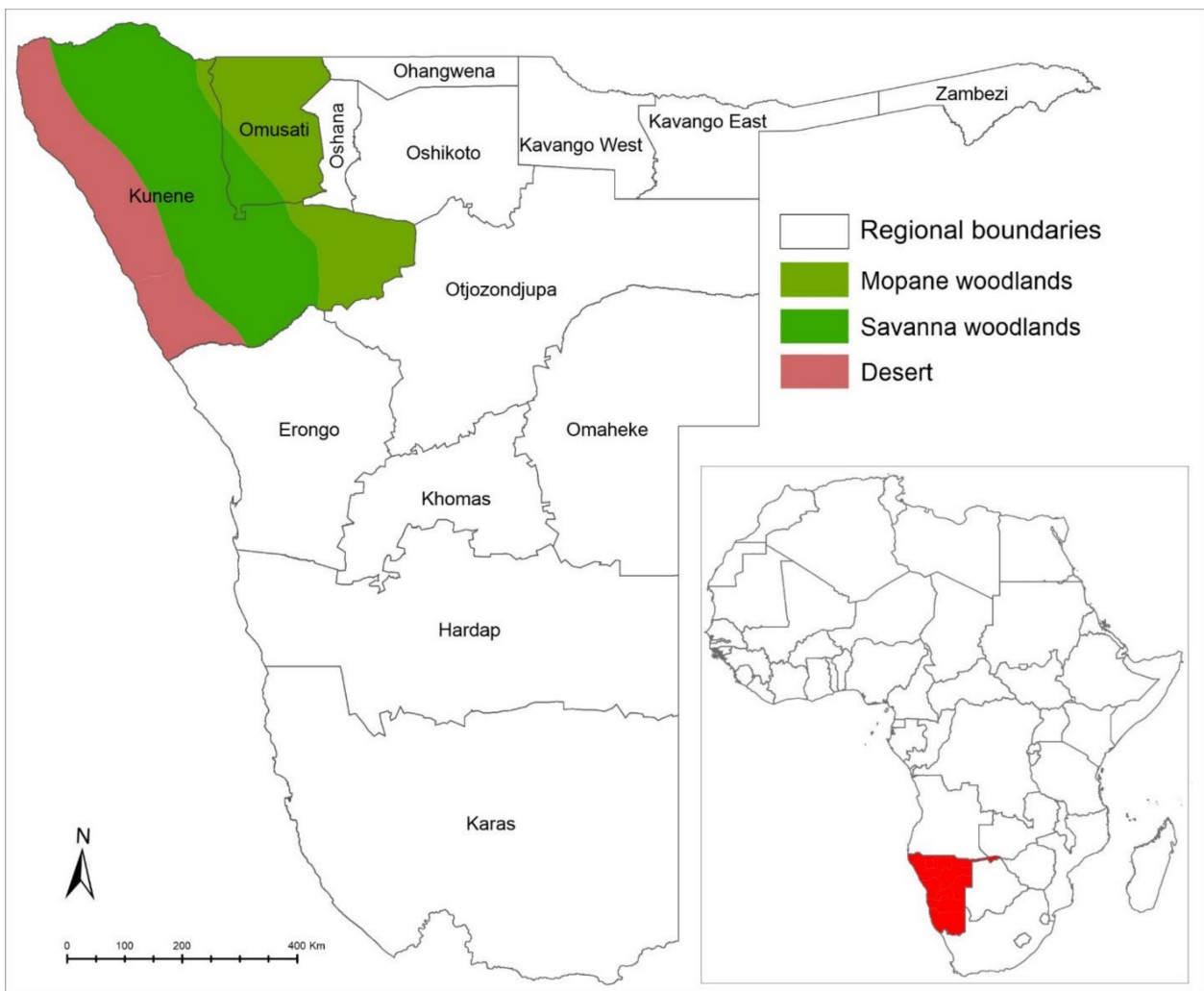

**Figure 1.** The location of the study area and key descriptions of the area. The red area is the location of Namibia on the map of Africa.

*2.2. Focal Species*

*C. mopane* belongs to the Fabaceae family [32]. The tree species can grow into big trees, ranging from 4 m to 18 m in height [32,40]. It is often present in alluvial, alkaline, and poorly drained soils, which it tolerates better than other species [32]. The bark has a rough texture that ranges from dark grey to blackish [34]. The heartwood is dark reddish to almost black, durable, rugged, and heavy [40]. The leaves are drooping and made up of two leaflets that look like butterfly wings. During the winter, the leaves are shed, and most trees are bare by the end of the season. *C. mopane* often forms pure stands of two distinct types [40]. On favorable sites, the stands are made up of tall trees referred to as 'cathedral mopane' but when the soil conditions are less favorable, the vegetation is referred to as 'mopane shrub' [53].

C. mopane vegetation is found in about eight countries in southern Africa, where many different ethnic groups live [33]. The distribution of *C. mopane* is best associated with low to moderate rainfall, high temperatures, low altitudes, and various soil types [43]. Hence, the *C. mopane* savanna covers a large area, extending south-western Angola and into Namibia, as far south as Brandberg mountain, the highest peak in Namibia [54] (Figure 2).

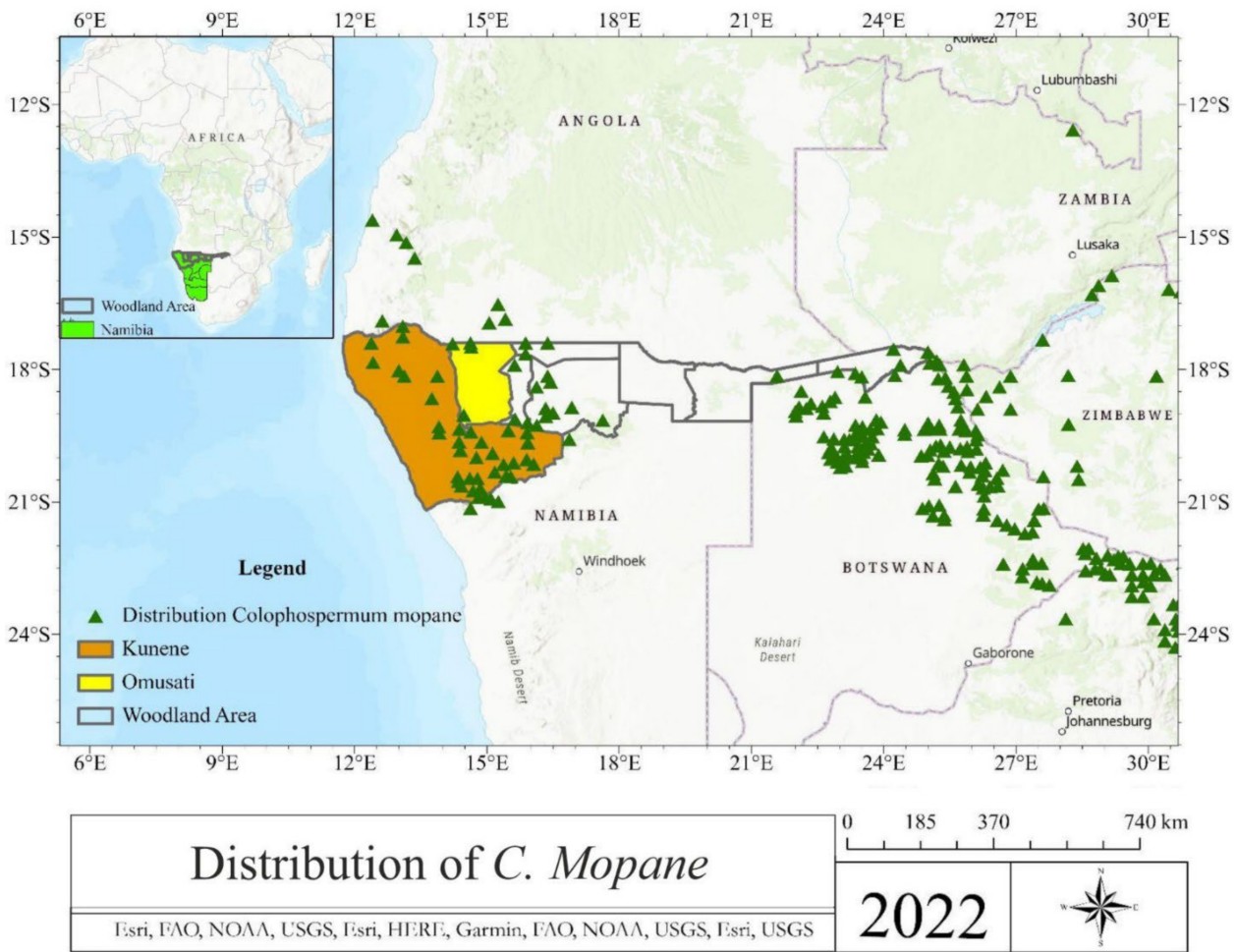

**Figure 2.** The distribution of *C. mopane*.

*C. mopane* prefers fine-grained sand and clay-loam sites formed from basalt, alluvial material, and lime [41]. The species is more competitive than any other species, mainly when the sites are periodically waterlogged and on solonetzic sites. *C. mopane* is predominantly found in low to moderate rainfall, high temperatures, low altitudes, and various soil types [32]. As a result, the northern part of Namibia has the highest concentration of *C. mopane*.

*C. mopane* is considered a natural resource and a source of income in the daily lives of local people [32,33]. It is the primary source of poles in dry regions because of its durability and availability, and is a source of fuelwood, droppers, rafters, and bark rope for subsistence use by local farmers in the most northern parts of Namibia [40]. The tree species is used for similar purposes in other parts of southern Africa [55]. Due to its dominance in the area [42,56], high economic values, and consequently high demands for multiple uses at the local community level, *C. mopane* was chosen as the focal species for this study.

### 2.3. C. mopane Products Data

One of the main challenges in forest ecosystems and forest resource utilization by local communities' research is a lack of data. As a result, we could only obtain data for *C. mopane*

products from 2011 to 2021. We focused on the five main products that local communities harvest from *C. mopane* in the area [42]. These products included firewood, poles, rafters, droppers, and roots. Local communities use such products for subsistence use (own use) and commercial purposes [42,45].

We gathered data for *C. mopane* products from harvesting permit record books through the Directorate of Forestry (DoF) of the Ministry of Environment, Forestry, and Tourism (MEFT). We used harvest permit books from all five DoF offices (Okahao, Onesi, Outapi Opuwo, and Tsandi) in the Omusati and Kunene regions. Data collected included the types of products (firewood, poles, droppers, rafters, and roots), uses, quantities, permit costs, and harvesting years. The five DoF offices formed the hotspots of our study. We collected data for 239 villages across the study area from these hotspots. Data included all harvesting permits of *C. mopane* from private and community forests from 2011–2021.

According to the pricing structure of the DoF, local communities pay between NAD 10–60 (Namibian dollars, equivalent to USD 0.67) depending on the type of permit (harvesting, transportation, and marketing) and use (commercial and own use/subsistence). The permit's validity differs with permits ranging from seven days to three months.

Furthermore, it is worth highlighting that due to the limited resources, the Directorate does not encourage the commercialization of forest products at the local level [57]. However, local communities sell forest products for commercial use in the following forms: poles, droppers, and rafters at NAD 10 per piece; and firewood and roots at NAD 12 per bundle [57].

*2.4. Data Analyses*

Estimations of the local weights of different wood products harvested by the local communities were developed by the authors of [57]. On average, a bundle of firewood/root is 0.013 tons, whereas poles are 0.0094 tons, droppers are 0.001 tons, and rafters are 0.001 tons per piece. To estimate the income received by the collector, we used the following formula:

(Total amount of collected wood products (in $m^3$) $\times$ market price/$m^3$). As the permit fee amount was negligible, we did not include it in the formula.

The continuous data were first checked for normality using the Kolmogorov–Smirnov normality test. The non-categorical and not-normally distributed data, such as harvest per product changes and potential benefit received by the respondents over time, were analyzed using the Kruskal–Wallis test, followed by the Bonferroni test for pairwise comparisons using 2011 as the reference year. We analyzed data using IBM SPSS version 26 (IBM Corp., Armonk, NY, USA). We entered, coded, and classified the data according to the two main uses, forest products and years of production. A *p*-value of less than 0.05 was deemed statistically significant in all analyses in this paper. According to the permit system of the DoF, forest products are recorded in different units. For example, there are pieces for poles, droppers, rafters, and tons for firewood and roots. Therefore, we standardized the units to tons (Table A1, Appendix A).

*2.5. Climate Trend Analyses*

We further performed climate trend analyses focusing on temperature and precipitation over the study period (2011–2021). The analyses aimed to provide a picture of how climate change affects forest ecosystems in various ways [58,59] and, consequently, harvesting patterns. However, the relationship between climate change and forest production is complex. It requires long-term data for all variables. That is why there is little previous research on this aspect in Namibia. Therefore, our analyses only give an overview of the changes in climate parameters (temperature and precipitation) to provide a baseline for future long-term analyses. We obtained temperature and precipitation data from the Southern African Science Service Centre for Climate Change and Adaptive Land Management (SASSCAL).

## 3. Results

### 3.1. Harvests per Use Types

The products were classified according to the two main use types, namely subsistence and commercial (Figure 3a,b).

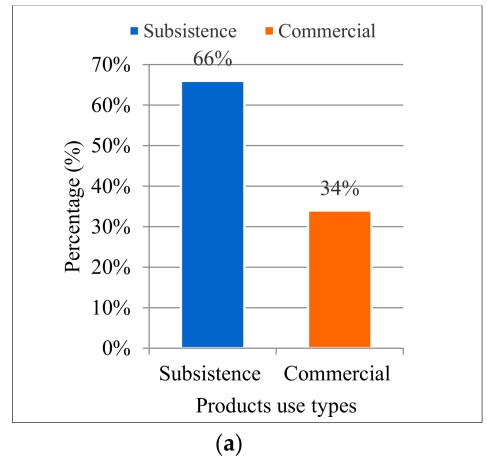 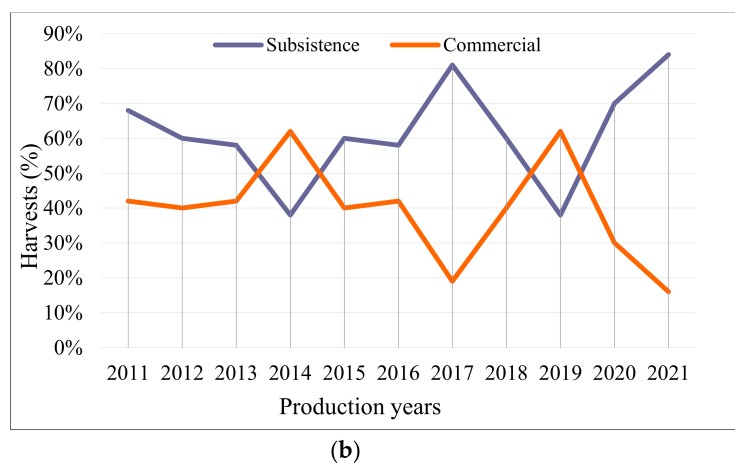

(**a**)  (**b**)

**Figure 3.** Product use types for *C. mopane* from 2011 to 2021; (**a**) percentage of two main types of use, and (**b**): trends in the harvests of products by use type from 2011–2021.

Our results further revealed that local communities mainly harvest *C. mopane* products for subsistence use (66%) (Figure 3a). However, there were also harvests for commercial products, which accounted for 34%.

Harvests of *C. mopane* products have fluctuated over the years (Figure 3b). Harvests for subsistence use were highest in the years 2021 (84%) and 2017 (81%). The lowest harvests for subsistence use were recorded with equal scores of 38% in the years 2014 and 2019. On the other hand, harvests for commercial use showed the highest same records of 62% in 2014 and 2019. Finally, we identified the lowest harvest records for commercial use in 2021 (15%).

### 3.2. Harvests per Products

There have been continuous harvests for multiple *C. mopane* products harvested annually between 2011 and 2021 in the Kunene and Omusati regions (Figure 4).

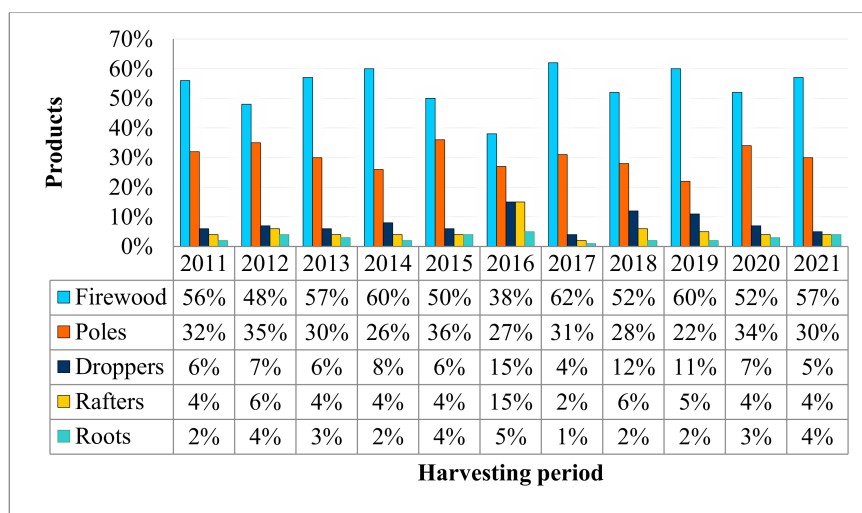

**Figure 4.** *C. mopane* products harvested per year.

The results showed that throughout the years (2011–2021), the highest harvests recorded were for firewood and poles. For example, in 2011, firewood harvests were at 56%, followed by poles (32%), and the lowest was roots (2%). Similar harvests were recorded in subsequent years. However, a slight decline was recorded in 2016 in the harvests of firewood (38%) and poles (26%). Conversely, a slight rise was recorded in the harvests for droppers (15%) and rafters (15%) in 2017. However, there was an increase in the harvests of the main products after 2016. For example, firewood comprised the highest record of 62%, while poles were at 31% in 2017.

Table A1 (Appendix A) shows changes in each product use from 2011 to 2021 and the comparison to the baseline year (2011). There was a significant increase in the production of firewood from 2011 to 2021 ($p < 0.001$), but the other years were similar to the reference year. Meanwhile, the production of droppers significantly increased in 2012 compared to 2011 ($p < 0.001$), while the other years were considered not statistically different.

### 3.3. Total Harvests

The proportion of the main product harvests showed that firewood and poles were the highest (Figure 5).

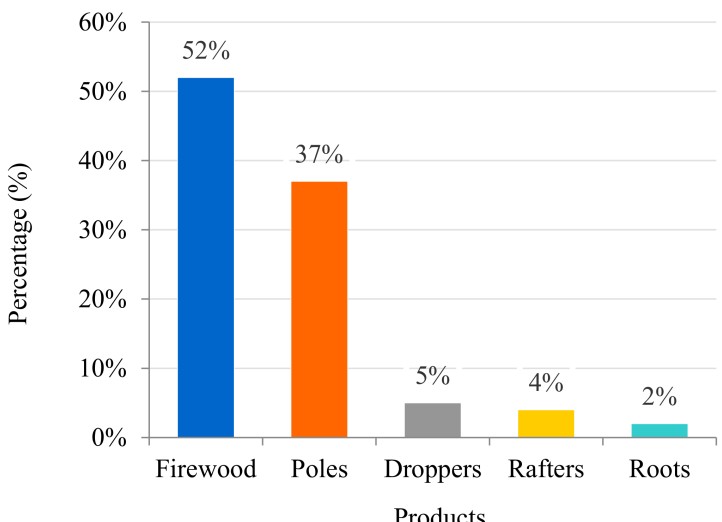

**Figure 5.** Total harvests of the primary forest products from *C. mopane*.

Our results showed that all five products were harvested throughout the study period, but in different quantities. Our total harvests analyses showed that firewood was significantly the highest forest product harvested from *C. mopane* (52%) during the entire study period. Secondly, poles (37%) were also considerably harvested over the study period. The least harvested products during the study period were droppers (5%), rafters (4%), and roots (2%).

### 3.4. Total Temperature and Precipitation Changes

Although changes in climate are best investigated over a long time, our analyses give an overview of how likely it is that climate change will affect forest production and the livelihood of local communities in *C. mopane* woodlands (Figure 6).

It is generally understood that climate change has various negative impacts on forest ecosystems, including forest fires and tree mortality caused by warming and frequencies of drought and flood occurrence [37,60,61]. We identified a negative correlation between changes in temperatures and precipitation, which can be perceived as a threat to the future of *C. mopane* ecosystems and production outputs.

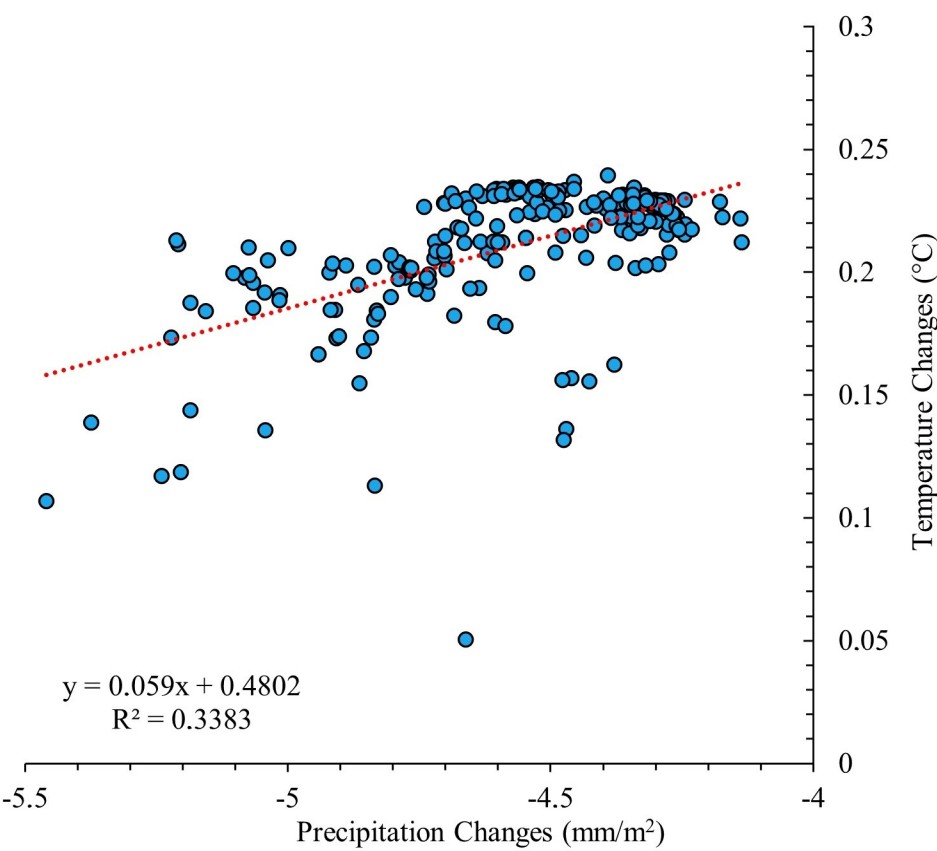

**Figure 6.** Changes in the amount of precipitation and temperature over the study period.

There was a maximum decrease in precipitation in the areas with a minimum increase in temperature across the study area. In general, the trends show that the temperature increased during the study area, whereas the precipitation decreased. A decrease in precipitation results in low forest productivity. For example, trees' growth will be stunted and pest and disease outbreak will worsen.

## 4. Discussion

Forest ecosystems play an essential role in the livelihoods of rural communities in southern Africa and other developing parts of the world, where forests are the vital elements of livelihood [62]. Nearly 2.9 billion people in low and middle-income countries cook and heat their homes by burning solid fuels, such as fuelwood [63].

*C. mopane* displays adaptive mechanisms to the dry conditions of Namibia. One of the adaptive features of the species is deep root systems [7,32]. As a result, this species can regenerate and grow despite the changing climatic conditions [64]. Hence, our results revealed that *C. mopane* woodlands provide and would potentially continue to provide goods and services for the livelihood of local communities, especially in the northern regions of Namibia (Figure 4). The main forest products identified were firewood for cooking, heating, and lighting and poles for construction.

Our results indicate that firewood is the northern area's most significant *C. mopane* product (Figure 5). Statistically, our results show that there was a significant increase in the production of firewood over the entire study period ($p < 0.001$). On the contrary, the production of other products has been fluctuating. For example, droppers significantly increased in 2012 compared to 2011 ($p < 0.001$) (Table A1, Appendix A).

The high firewood consumption is because the area is primarily rural, where most residents depend on firewood for lighting and heating [45,56,65]. For example, most residents in the northern regions of Namibia use firewood primarily for cooking [66]. Our results coincide with the literature which reveals that many African countries have high

demands for fuelwood consumption. For example, Ethiopia (93%), Nigeria (80%), and the Democratic Republic of the Congo (74%) are the leading countries in the use of firewood at the local level [67]. In South Africa, recent research shows that fuelwood is still used to some extent by 96% of rural households [68,69].

Our results further reveal that local communities mostly use *C. mopane* products for subsistence (Figure 3). These indications are in alignment with the mandate of DoF regulations. Due to climatic conditions, coupled with limited forest resources, the management principles of forest ecosystems of Namibia do not encourage the commercialization of forest resources at the community level [45,70]. Through these forestry management principles, forest managers, in collaboration with the relevant stakeholders, exercise a sustainable management approach to curb the possible depletion of resources [45]. Such management principles also consider deforestation and forest degradation alarming, aggravating climate change [71]. At the same time, it is worth highlighting that management approaches that exclude local communities from forest resources invite illegal activities from local communities close to forest ecosystems [72,73]. Therefore, it is essential to promote the sustainable utilization of forest resources.

Additionally, it is crucial to emphasize that the continuous utilization of forest products in a changing climate depend on the effectiveness of collective adaptation and sustainable forest management approaches [30]. Effective law enforcement is one strategy to harmonize adaptation and sustainably manage forest ecosystems to avoid illegal operations, such as unlawful harvesting. Afforestation and reforestation are other approaches that need strengthening to sustain forest ecosystems [70]. Such forest management approaches are essential in *C. mopane* woodlands, given species' the ability to survive the area's harsh climatic conditions. In addition, silvicultural activities against forest disturbances, such as forest fires, pests, and diseases, also need strengthening [30,37]. However, there is limited scientific research in the context of climate change, forest ecosystems, and projected disturbances in Namibia.

Our climate change analyses showed that there was a maximum decrease in precipitation in the areas with a minimum increase in temperature across the study area (Figure 6). The temperature increased over the study area, whereas the precipitation decreased. As generally perceived, a decrease in precipitation has a negative impact on forest productivity which manifests in various ways. For example, trees' growth will be stunted and pest and disease outbreak will worsen. Increasing temperatures will also alter the functions of the ecological systems of the forest [61]. However, more research is urgently needed to investigate this phenomenon from this angle in forest ecosystems in northern Namibia.

Food security is another crucial aspect of discussions on forest ecosystem services in a changing climate [74]. For example, during low rainfall, local communities do not produce good yields from other activities that support their livelihoods. In the case of the northern regions of Namibia, most rural communities depend on livestock and crop farming [43,75,76]. However, due to periodic erratic rainfalls and severe droughts, local communities diversify their livelihoods to off-farm activities [43], including using forest resources for income generation to support their livelihoods [42]. As a result, local communities use of forest resources will worsen during severe droughts.

Finally, we identified that this type of study requires long-term climate data, particularly precipitation and temperature trends concerning growth and mortality rates, to determine the status of the forest ecosystems. Unfortunately, such data are lacking on Namibia's forest ecosystems and climate change. We also noticed that the commercialization of forest products needs intensive investigation and monitoring to evaluate the value of products at the local community level.

## 5. Conclusions

Our analyses showed that local communities harvested five main products from *C. mopane*, namely firewood, poles, droppers, rafters, and roots. Firewood was the primary forest product, followed by poles and droppers. Total harvests appeared to fluctuate over

the years. Therefore, our results suggest that forest ecosystems will continue to potentially benefit the livelihoods of local communities in northern Namibia despite the changing climate. However, there is a need for predictive analyses to estimate the rate at which forest ecosystems will potentially continue to support local livelihoods in correlation with the changes in climate parameters, such as precipitation and temperature.

An increase in the temperature, and a decrease as identified over the study period, will potentially result in the circumstances, such as severe droughts, local communities are most likely to exert considerable pressure on forest ecosystems for production outputs. This could be one of the many ways climate change contributes to the degradation of forest ecosystems from the perspective of production outputs. Unfortunately, Namibia lacks a robust national system that provides data for forest products at the rural community level. There is also a lack of spatially extensive climate data. Therefore, there is limited access to digitized data at the country level. Therefore, a national system is needed to provide forest ecosystem service information to the public through DoF offices countrywide.

In addition, to help the country produce effective adaption and mitigation measures against climate change, it will be ideal if the MEFT and relevant stakeholders establish a climate database system that records forest products, weather, and climate data. We further propose that future studies focus on predictive analyses of the impacts of climate change on the status of forest ecosystem services in Namibia. More specifically, future research needs to assess extreme weather events, such as forest fires, droughts, floods, and other climate-related hazards, that affect goods and services provided by forest ecosystems in the northern regions and the entire country. Furthermore, this study has the potential to serve as a forest ecosystem management tool for the forestry sector in Namibia. In the same view, in collaboration with local communities, forestry managers can draw insights from this study to formulate sustainable forest ecosystem management strategies to ensure that *C. mopane* ecosystems continue to provide socio-economic benefits for local communities in a changing climate. Finally, the utilization and commercialization of forest products at the local community level should be closely monitored through a robust database system to ensure sustainability and adaptation to a changing climate.

**Author Contributions:** Conceptualization A.N. and M.H.; methodology, A.N., A.A., A.K. and D.P.; software, A.A. and D.P.; validation, M.H.; formal analysis, A.N., A.A. and D.P.; investigation, A.N., A.A., A.K. and D.P.; resources, A.N. and M.H.; data curation, A.N. and A.K.; writing—original draft preparation, A.N.; writing—review and editing, A.N.; visualization, A.N., A.A. and D.P.; supervision, M.H.; project administration, M.H.; funding acquisition, M.H. All authors have read and agreed to the published version of the manuscript.

**Funding:** This work was supported by the Operational Program Research, Development, and Education, the Ministry of Education, Youth, and Sports of the Czech Republic grant No. CZ.02.1.01/0.0/0.0/16_019/0000803.

**Institutional Review Board Statement:** The study was conducted in accordance with the ethical conduct of the National Commission on Research Science and Technology (NCRST), issued on 13 December 2022, and the ethical clearance by the Ministry of Environment, Forestry and Tourism (MEFT) of Namibia, and approved by the Directorate of Forestry (DoF), dated 18 August 2022.

**Informed Consent Statement:** Written informed consent has been obtained from the Ministry of Environment, Forestry, and Tourism of Namibia and the National Commission on Research Science and Technology (NCRST) to publish this paper.

**Data Availability Statement:** All data relevant to the study are included in the article.

**Acknowledgments:** We thank individual forestry officials from the Omusati and Kunene regions who assisted with data collection from various DoF offices that participated in this study. Finally, we are grateful to the anonymous reviewers for their constructive reviews.

**Conflicts of Interest:** The authors declare that they have no known competing financial interests or personal relationships that could have appeared to influence the work reported in this paper.

## Appendix A

**Table A1.** Changes in wood product volume collection compared to the referral year (2021) [1].

| Wood Products | Referral Year | Observed Year | | | | | | | | | |
|---|---|---|---|---|---|---|---|---|---|---|---|
| | 2011 | 2012 | 2013 | 2014 | 2015 | 2016 | 2017 | 2018 | 2019 | 2020 | 2021 |
| **Poles (N = 1222)** | | | | | | | | | | | |
| Volume, m³ | 40,000 (1;440,000) | 80,000 ** (3;200,000) | 80,000 ** (10;500,000) | 55 ** (6;240,000) | 100 ** (1;1,000,000) | 30 ** (1;200,000) | 120 ** (1;600,000) | 100,000 ** (30;150,000) | 55,000 (8;1200,000) | 500,000 * (1000;45,000,000) | 33,000 (1000;200,000) |
| Income, ×1000 NAD | 20,000 (1;220,000) | 40,000* (2;100,000) | 40,000 ** (5;250,000) | 27,500 ** (3;120,000) | 50,000 ** (5;500,000) | 15,000 ** (1;100,000) | 30,000 ** (1;300,000) | 50,000 ** (15;75000) | 27,500 (4;575000) | 25,000 * (500;22000,000) | 16,500 (500;100,000) |
| **Firewood (N = 1723)** | | | | | | | | | | | |
| Volume, m³ | 1000 (1;7000) | 100 ** (1;2000) | 100 ** (1;2000) | 60 ** (4;44563) | 60 ** (1;2000) | 4 ** (1;150,000) | 100 ** (1;10,000) | 100 ** (6;80,000) | 1000 * (8;10,000) | 1000 * (20;145000) | 1000 ** (5;86000) |
| Income, ×1000 NAD | 500 (0.4;3500) | 50 ** (5;1000) | 50 ** (5;3500) | 30 ** (2;22,282) | 30 ** (1;1000) | 2 ** (1;75000 | 50 ** (5;5000) | 500 ** (3;40,000) | 500 * (4;5000) | 500 * (10;72,500) | 500 ** (3;43,000) |
| **Droppers (N = 190)** | | | | | | | | | | | |
| Volume, m³ | 40 (3;240,000) | 14,000 (60,000;200,000) | 120,000 ** (30;340,000) | 50 (10;150) | 81 (1;600,000) | 300,000 ** (1;334,000) | 120 (50;100,000) | 100 (30;200) | 100,000 * (80,000;250,000) | 100,000 ** (90,000;200,000) | 175,000 ** (2000;500,000) |
| Income, ×1000 NAD | 3 (0.2;16,216) | 9460 (4054;13,5135) | 8108 ** (1;22,973) | 3 (1;10) | 6 (0.1;40,541) | 20270 ** (0.1;22,566) | 8 (3;6757) | 7 (2;15.5) | 6757 * (5405;16,892) | 6757 ** (1351;60,811) | 11824 ** (135;33,784) |
| **Rafters and Roots** | | | | | | | | | | | |
| Volume, m³ | 100500 (1000;200,000) | 80,000 (24,000;380,000) | 100,000 (1000;400,000) | 80,010 (20;160,000) | 80,000 (25000;110,000) | 80,000 (6;150,000) | 99,000 (2;1900,000) | 75,000 (2000;200,000) | 90,000 (40,000;840,000) | 70,000 (1000;200,000) | 40,000 (1000;250,000) |
| Income, ×1000 NAD | 6965 (417;13,514) | 5405 (1622;25,676) | 6757 (417;27,027) | 5410 (1689;7432) | 5405 (3;10,135) | 6689 (1;128,378) | 5068 (676;13,514) | 6081 (2703;56,757) | 5070 (677;13,513) | 4230 (417;135,135) | 2703 (68;16,892) |

[1] Volume and income were presented as median (min, max), Group comparisons were analyzed using the Kruskal–Wallis test, continued by the Bonferroni test for pairwise comparison with the referral year (2021). Income was calculated as volume/piece (in m³) × market price of wood product (NAD)/piece (in m³). * = significant level at $p < 0.05$ with the referral year (2021); ** = significant level at $p < 0.001$ with the referral year (2021).

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
