# Peer review of "Socio-Economic Benefits of Colophospermum mopane in a Changing Climate in Northern Namibia"

_forests, doi:10.3390/f14020290_

Round 1
Reviewer 1 Report (New Reviewer)
Initially hesitated whether to reject the paper or to give it a chance to go further and I decided anyway to give a chance to the authors, to improve the paper. Thematic is interesting and if the paper is well prepared, the results could be useful for the African local communities. If this manuscript is going to become a real paper, I sincerely recommend authors to revise it. My remarks are as follows:
1. The manuscript is titled “Socio-economic benefits of…”. In the paper is nothing about the social and economic benefits for a society, even being a local community. The utilization of forests in particular specie, measured in precentage is barely connected to the social benefits and a little bit more to the economic, but under some constraints. I think the Introduction is OK, but let’s start with the methodology.
2. In the methodology are included just some indexes of the harvesting. This is not sufficient at all. At first authors have to valuate the production. Afterward they have to describe the type of incomes the production provide to the society. After the income is described has to be described the connection between these incomes and the social benefits. For example if the ecosystem go bad, what could be the circumstances to the welfare of the local communities. It has to be described better the benefit of the C. mopane specie in expense of other species in the local forests.
3. The discussion part should follow the results and not to describe something that has not been evaluated in the results part.
I wish the authors good luck with the manuscript
Author Response
Dear reviewer 1,
Thank you for your constructive comments. All the comments and observations were helpful for the improvement of the paper. We have learned a lot from you and sincerely hope our improvements will satisfy you. If you have more comments, we will do our best to attend to them. We appreciate your support.
Best regards,
Andreas

Reviewer 2 Report (New Reviewer)
Thank you, editor and authors, for providing an opportunity to read this manuscript. The information presented in the manuscript is important for coping with the impact of climate change. The paper is well-written. However, a substantial improvement is necessary for the methods and results section.
Comments:
2.3. Product data: Lack of detailed description of data collection. How had the data been collected? What type of data was collected from the DoF? Were those data accounting for all the harvested quantity? Were there any leakages (illegal harvesting)? For instance, local people can collect C. mopane without approval; how did the authors consider that?
I am just curious that where that C. mopane has been harvested, either from Community-based forest management or any national forests?
Lack of statistical analysis: The authors presented results in graphs and figures, which is good but not enough. I, therefore, suggest authors to carry out some statistical tests. At least a few tests to see statistically significant differences among the product types. In addition, a trend analysis can be done with respect to temperature and precipitation data. It provides a picture of how climate change affects the products' harvest.
Figure 4: Is it suitable to show in the web diagram?
Lines 247-253: Is this study suitable to compare with the study in China?
Specific Comments:
- Introduction: Please present the study objective explicitly.
- Line 142: Be sure to use either “Four” or “Five”
Author Response
Dear reviewer 2,
Thank you for your constructive comments. All the comments and observations were helpful for the improvement of the paper. We have learned a lot from you and sincerely hope our improvements will satisfy you. If you have more comments, we will do our best to attend to them. We appreciate your support.
Best regards,
Andreas

Reviewer 3 Report (New Reviewer)
The manuscript deals with the evaluation and analysis of the socio-economic benefits of forest goods and services in changing climate by focusing on the forest products derived from the species Colophospermum mopane in northern Namibia. In general, the manuscript is well-written, structured and informative, but still needs certain improvements before acceptance for publication in the Forests Journal. Please, see below my comments on your work:
In general, the title (lines 2-3), the abstract (lines 9 to 20) and the keywords (lines 22-22) correspond to the title, aims and objectives of the manuscript. The abstract is well-written and informative, and contains the main findings of the article.
Lines 47-54: there are some unnecessary repetitions, please revise accordingly.
Overall, the Introduction part is well written and informative, and provides relevant information and references on the research topic.
The Materials and Methods section is detailed and well-written.
Line 163: Figure 3 is not of very good quality, it would be better to provide a higher-quality graphic. Please provide captions for figure 3 a) and b).
Line 158: This section should be named “Results”, not “Results and discussion”. The discussion is given in section 4 of the manuscript.
The results are presented rather vague and simple, the text just duplicated what has already been given in the figures.
The Discussion section is rather general, and not directly related to the scope of the article.
The Conclusion part (lines 254-278) reflects the main findings of the manuscript. In addition to the potential of future studies, I would recommend adding also the practical application of your results for the forestry sector of the country.
The References cited are appropriate to the topic of the manuscript.
Author Response
Dear reviewer 3,
Thank you for your constructive comments. All the comments and observations were helpful for the improvement of the paper. We have learned much from you and sincerely hope our improvements will satisfy you. If you have more comments, we will do our best to attend to them. We appreciate your support.
Best regards,
Andreas

Round 2
Reviewer 2 Report (New Reviewer)
Thank you, authors, for revising the manuscript. I greatly appreciate your effort in improving the manuscript. However, a few things need to be considered before making it publishable in the journal.
1. 2.3 product data: It needs to be improved, as I said in the reviewer round 1. Be advised to explicitly mention units of quantities, cost per kgs or tons, etc. Also, at least, please mention it in the study limitation section or add a few sentences in the last part of the conclusion about the data collection (or study) limitations.
2. Figure 5: I suggest using another type of figure to show it. Because the sum of five items is 100%, these are mutually exclusive components. So, this spider web diagram is similar to that pie chart.
Author Response
Dear reviewer 2,
Thank you for your constructive comments. All the comments and observations were helpful for the improvement of the paper. We have learned much from you and sincerely hope our improvements will satisfy you. If you have more comments, we will do our best to attend to them. We appreciate your support and hope to have you on board in our next papers again.
Best regards,
Andreas

Reviewer 3 Report (New Reviewer)
The authors have greatly improved their manuscript, based on the reviewers' comments. I have no further remarks, with the exception of one thing:
Line 188: the caption of Figure 3 should be below the respective figure, now it is placed at the beginning of the page, which is not appropriate, please revise.
Author Response
Dear reviewer 3,
Thank you for your constructive comments. All the comments and observations were helpful for the improvement of the paper. We have learned much from you and sincerely hope our improvements will satisfy you. If you have more comments, we will do our best to attend to them. We appreciate your support and hope to have you on board in our next papers again.
Best regards,
Andreas

This manuscript is a resubmission of an earlier submission. The following is a list of the peer review reports and author responses from that submission.
Round 1
Reviewer 1 Report
Thank you for the opportunity to review this paper which reports on a spatial modelling analysis of rainfall and temperature in norther Namibia and attempts to correlate that to the quantity of several products harvested from C. mopane trees in the study area. The authors conclude that there has been a slight increase in temperatures over the last decade and a corresponding slight decline in rainfall. They subsequently conclude that the quantity of several mopane products are then influenced by these changes.
Generally, I found the paper easy to read and reasonably well situated in relative debates and literature. Yet I think it have several significant shortfalls which make me doubt its suitability for publication in Forests. I outline these below.
1. A major question relates to the validity of the data on C. mopane use. The authors describe that use data were accessed from the permits issued at five permitting offices for 298 villages (L218). The validity and completeness of the permit data is never assessed, questioned nor reported. Based on my own work in southern Africa and SE Asia, I find it hard to believe that people will travel tens of kilometres to the closest permit office every time they wish to harvest firewood. Energy is a basic human need, and in that region people go out 2-4 times a week to collect firewood to cook. Having to go via a permit office will cost them a taxi fare, or several hours of walking/donkey cart to get there. Therefore, it is highly unlikely that they actually do so. Indeed, in the conclusions (L446) the authors express concern about the lack of “an effective national system that offers data for production outputs”, which seems to corroborate my concern. Therefore, I am sceptical that the permit data actually reflect reality on the ground. I need some reassurance that the data are good enough to be used in the analysis.
2. A concern is the absence of even basic statistical analysis of the data or displays of standard deviations on the graphs. The absence of any statistical analysis is necessary to validate (or not) the claimed differences and trends. For example, most of the average values in Figure 8 do not differ by large magnitudes across the precipitation and temperature classes. If these small differences are not validated via suitable stats analyses, then the whole conclusion of the paper is invalidated. In fact, the authors state on L381 that the “correlations seemed insignificant” ….. but changes in precipitation influence harvests “to some extent”. That raises the question, ‘to what extent’ and what is the statistical significance of the relationship? How much of the co-variance is accounted for by the relationship?
3. The absence of means and standard deviations is also telling with respect to the claimed changes in precipitation and rainfall. The absolute values of change are miniscule in relation to the long-term means. It is well known that the variability in rainfall increases as mean annual amounts decrease; in other words arid and semi-arid regions have high coefficients of variation in rainfall. Thus, a change of 4-5 mm p.a in the mean as considered in this study will be dwarfed by the normal variation in annual rainfall in that region. How did the authors tease out such a small change in such a noisy data set, especially over such a small time period? The same applies with temperatures, where a change of 0.20 C is unlikely to have been noticed by anybody in the region, when maximums can frequently exceed 400 C, i.e. 0.5 %, which is well within the ‘normal’ variation for the area.
4. The authors conclude that the alleged greater vegetation cover in the central region “played a significant role in preserving the perception despite the high increase in temperature” (L247). This is further explained in the discussion (L336+; and L416). I have three concerns about this conclusion. The first is that no data or information are presented to show that the central region does actually have a higher vegetation cover than the eastern or western portions. If this is central to any explanations of the patterns in the climate data then some real results regarding measures of vegetation cover need to be presented.
Secondly, vegetation might “preserve” precipitation that falls, but in arid environments it does not result in higher precipitation (the cited reference is not from an arid area). But the data displayed in Figure 4 are the change in precipitation that fell, not the moisture that was retained. Therefore, this explanation is spurious.
Thirdly, the whole of the paragraph starting on L413 is mostly speculation. The authors say that the area would “receive significant rainfall” and that “the ground will be saturated with water”. But as already argued in point 3 above, the percentage change the authors suggest is negligible and is well within the normal variation for an arid system. The comments about what it means for regeneration and then harvests is also speculation because they have not considered possible counter-effects of higher rainfall such as increased tree competition, or fire frequency and severity due to increased fuel-loads. In fact, the statement on L414 is directly contradicted on L433. The first says that increased temperature will increase precipitation, but the second says it will decrease it.
5. I did not understand how or why the use of C. mopane products was reported in percentages. In particular, how can Figure 6 display all products as percentages when the units of the empirical measures for different C. mopane products different (tons and pieces).
6. On L350 the authors recommend action “against forest fires”. This is very problematic because it is contrary to the wealth of literature showing that fires are a natural and integral ecosystem agent in semi-arid savannas and woodlands around the world, including C. mopane woodlands. There is so much literature in this regard that I do not need to point any specific studies.
7. I find the statement on L361 about links between commercial use of forest products and the likelihood for overuse a problematic generalization. The sustainability of use depends on a large number of local- and higher-scale consideration. A key one is tenure and governance – see for example the work of Elinor Ostrom and colleagues. There are many instances where commercial use is and can be sustainable (see for example the review by Stanley et al. (2012) or the book edited by Shackleton et al. (2015)).
8. Reference is made to a couple of tables. However, the version of the paper I had access to had no tables nor any supplementary materials (where the tables might be?).
9. A small comment is that on L116 the authors state that 302 ha of tree cover in the region was lost between 2001 and 2018. That equates to 17.8 ha per year. In an area of approx 140,130 km2 (17% of Namibia), that means a loss of 0.00013 % p.a. In relation to deforestation statistics from around the world this is laughable. Unless of course, the authors have reported the units incorrectly?
10. Section 3 covers Results and Discussion, yet Section 4 covers Discussion. Having the discussion in two places is both confusing and unconventional. The paper would read better if the discussion was in one place and not two.
11. There are multiple instances of direct repetition in the paper that need to be deleted.
12. Figure 2: the small side graphs are illegible, even on the online version.
Author Response
Authors' Responses to Reviewer's Comments (Reviewer 1)
English language and style are fine/minor spell check required |
Thank you, reviewer, for the comment. We have improved the language accordingly. |
Are the methods adequately described? – Must be improved |
Thank you, reviewer, for the constructive comment. We have improved the methods according to the revised topic. |
Are the results clearly presented? - Must be improved |
Thank you, reviewer, for the constructive comment. We have improved the results according to the revised topic. |
Are the conclusions supported by the results? – Must be improved |
Thank you, reviewer, for the constructive comment. We have improved the conclusions according to the revised topic. |
Thank you for the opportunity to review this paper which reports on a spatial modelling analysis of rainfall and temperature in northern Namibia and attempts to correlate that to the quantity of several products harvested from C. mopane trees in the study area. The authors conclude that there has been a slight increase in temperatures over the last decade and a corresponding slight decline in rainfall. They subsequently conclude that the quantity of several mopane products are then influenced by these changes. |
Dear reviewer,
Thank you for your observations and constructive comments. Please be informed that we changed the aim of the idea. We changed the entire title to “Socio-economic benefits of Colophospermum mopane in a changing climate in northern Namibia” |
Generally, I found the paper easy to read and reasonably well situated in relative debates and literature. Yet I think it have several significant shortfalls which make me doubt its suitability for publication in Forests. I outline these below. |
Dear reviewer, Thank you for your observations and constructive comments. Please be informed that we changed the aim of the idea. We changed the entire title to “Socio-economic benefits of Colophospermum mopane in a changing climate in northern Namibia” |
A major question relates to the validity of the data on C. mopane use. The authors describe that use data were accessed from the permits issued at five permitting offices for 298 villages (L218). The validity and completeness of the permit data is never assessed, questioned nor reported. Based on my own work in southern Africa and SE Asia, I find it hard to believe that people will travel tens of kilometres to the closest permit office every time they wish to harvest firewood. Energy is a basic human need, and in that region people go out 2-4 times a week to collect firewood to cook. Having to go via a permit office will cost them a taxi fare, or several hours of walking/donkey cart to get there. Therefore, it is highly unlikely that they actually do so. Indeed, in the conclusions (L446) the authors express concern about the lack of “an effective national system that offers data for production outputs”, which seems to corroborate my concern. Therefore, I am sceptical that the permit data actually reflect reality on the ground. I need some reassurance that the data are good enough to be used in the analysis. |
Thank you for the comments. However, we would like to point out that the forestry system of Namibia allows local communities to have access to forest resources including firewood. In most cases, local communities are required to obtain harvesting permits from forestry offices. Therefore, we have confidence in the data obtained through harvesting permit books/records. In the same view, this was the only way to obtain long-term records (2011-2021). The other way to acquire such information would be a survey, which is what we are currently running for the upcoming papers. However, it is not possible to obtain accurate information about the quantities harvested over the past ten years through a survey. If in doubt, we can provide the permit records from where we extracted the data. |
A concern is the absence of even basic statistical analysis of the data or displays of standard deviations on the graphs. The absence of any statistical analysis is necessary to validate (or not) the claimed differences and trends. For example, most of the average values in Figure 8 do not differ by large magnitudes across the precipitation and temperature classes. If these small differences are not validated via suitable stats analyses, then the whole conclusion of the paper is invalidated. In fact, the authors state on L381 that the “correlations seemed insignificant” ….. but changes in precipitation influence harvests “to some extent”. That raises the question, ‘to what extent’ and what is the statistical significance of the relationship? How much of the co-variance is accounted for by the relationship? |
Due to the complexity of the subject of climate change, we did not use any climate change data. This is because there is no direct relationship between climate change and forest products. However, we made suggestions about how to address this matter.
We described the statistical analysis as follows: We performed data analysis in Microsoft® Excel® for Microsoft 365 MSO (Version 2211 Build 16.0.15831.20098) 64-bit. We entered, coded, and classified the data according to the two main uses, forest products and years of production. According to the permit system of DoF, forest products are recorded in different units. For example, there are pieces for poles, droppers, rafters, and tons for firewood and roots. Therefore, we standardized our results to the percentage (%). |
The absence of means and standard deviations is also telling with respect to the claimed changes in precipitation and rainfall. The absolute values of change are miniscule in relation to the long-term means. It is well known that the variability in rainfall increases as mean annual amounts decrease; in other words arid and semi-arid regions have high coefficients of variation in rainfall. Thus, a change of 4-5 mm p.a in the mean as considered in this study will be dwarfed by the normal variation in annual rainfall in that region. How did the authors tease out such a small change in such a noisy data set, especially over such a small time period? The same applies with temperatures, where a change of 0.20 C is unlikely to have been noticed by anybody in the region, when maximums can frequently exceed 400 C, i.e. 0.5 %, which is well within the ‘normal’ variation for the area. |
Due to the complexity of the subject of climate change, we did not use any climate change data. This is because there is no direct relationship between climate change and forest products. However, we made suggestions about how to address this matter.
We described the statistical analysis as follows: We performed data analysis in Microsoft® Excel® for Microsoft 365 MSO (Version 2211 Build 16.0.15831.20098) 64-bit. We entered, coded, and classified the data according to the two main uses, forest products and years of production. According to the permit system of DoF, forest products are recorded in different units. For example, there are pieces for poles, droppers, rafters, and tons for firewood and roots. Therefore, we standardized our results to the percentage (%). |
The authors conclude that the alleged greater vegetation cover in the central region “played a significant role in preserving the perception despite the high increase in temperature” (L247). This is further explained in the discussion (L336+; and L416). I have three concerns about this conclusion. The first is that no data or information are presented to show that the central region does actually have a higher vegetation cover than the eastern or western portions. If this is central to any explanations of the patterns in the climate data then some real results regarding measures of vegetation cover need to be presented. |
Due to the complexity of the subject of climate change, we did not use any climate change data. This is because there is no direct relationship between climate change and forest products. However, we made suggestions about how to address this matter.
Figure 8 has been removed.
|
Secondly, vegetation might “preserve” precipitation that falls, but in arid environments it does not result in higher precipitation (the cited reference is not from an arid area). But the data displayed in Figure 4 are the change in precipitation that fell, not the moisture that was retained. Therefore, this explanation is spurious. |
Due to the complexity of the subject of climate change, we did not use any climate change data. This is because there is no direct relationship between climate change and forest products. However, we made suggestions about how to address this matter.
Figure 4. Long-term changes in precipitation (mm/m2) in northern Namibia, has been removed. |
Thirdly, the whole of the paragraph starting on L413 is mostly speculation. The authors say that the area would “receive significant rainfall” and that “the ground will be saturated with water”. But as already argued in point 3 above, the percentage change the authors suggest is negligible and is well within the normal variation for an arid system. The comments about what it means for regeneration and then harvests is also speculation because they have not considered possible counter-effects of higher rainfall such as increased tree competition, or fire frequency and severity due to increased fuel-loads. In fact, the statement on L414 is directly contradicted on L433. The first says that increased temperature will increase precipitation, but the second says it will decrease it. |
Dear reviewer,
Thank you for your constructive observation and comments. That part has been completely removed due to the changes in the topic and research aim. |
I did not understand how or why the use of C. mopane products was reported in percentages. In particular, how can Figure 6 display all products as percentages when the units of the empirical measures for different C. mopane products different (tons and pieces). |
According to the permit system of DoF, forest products are recorded in different units. For example, there are pieces for poles, droppers, rafters, and tons for firewood and roots. Therefore, we standardized our results to the percentage (%). |
On L350 the authors recommend action “against forest fires”. This is very problematic because it is contrary to the wealth of literature showing that fires are a natural and integral ecosystem agent in semi-arid savannas and woodlands around the world, including C. mopane woodlands. There is so much literature in this regard that I do not need to point any specific studies. |
We changed the entire title to “Socio-economic benefits of Colophospermum mopane in a changing climate in northern Namibia” This implies that our focus is now on the assessments of the socio-economic benefits of C. mopane in northern Namibia during the changing climate. Due to the complexity of the subject of climate change, we did not use any climate change data. This is because there is no direct relationship between climate change and forest products. However, we made suggestions about how to address this matter. |
I find the statement on L361 about links between commercial use of forest products and the likelihood for overuse a problematic generalization. The sustainability of use depends on a large number of local- and higher-scale consideration. A key one is tenure and governance – see for example the work of Elinor Ostrom and colleagues. There are many instances where commercial use is and can be sustainable (see for example the review by Stanley et al. (2012) or the book edited by Shackleton et al. (2015)). |
Dear reviewer, thank you for your comments. Due to the changes in topic, we addressed this point accordingly under discussion and conclusions. |
Reference is made to a couple of tables. However, the version of the paper I had access to had no tables nor any supplementary materials (where the tables might be?). |
Dear reviewer, thank you for your comments. Due to the changes in topic, we addressed this point accordingly under discussion and conclusions. Our discussions are now limited to the uses of forest products at the local rural areas level in Namibia, supported by literature from other developing countries. |
A small comment is that on L116 the authors state that 302 ha of tree cover in the region was lost between 2001 and 2018. That equates to 17.8 ha per year. In an area of approx 140,130 km2 (17% of Namibia), that means a loss of 0.00013 % p.a. In relation to deforestation statistics from around the world this is laughable. Unless of course, the authors have reported the units incorrectly? |
Dear reviewer, thank you for your constructive comment. We believe that this statement might sound shocking. However, we must point out that Namibia is a highly deforested country. Especially in the northern region which is the most populated area. The high demands for land for agricultural activities combined with illegal harvesting all lead to high deforestation. |
Section 3 covers Results and Discussion, yet Section 4 covers Discussion. Having the discussion in two places is both confusing and unconventional. The paper would read better if the discussion was in one place and not two. |
Dear reviewer, thank you for your comments. We have prepared these two sections according to the journal template. |
There are multiple instances of direct repetition in the paper that need to be deleted. |
Thank you for this constructive comment. We totally agree and have removed this information. |
Figure 2: the small side graphs are illegible, even on the online version. |
We have removed Figure 2 according to the new general aim of the manuscript. We considered it irrelevant. |
